# Deciphering the Genetic Variation: A Comparative Analysis of Parental and Attenuated Strains of the QXL87 Vaccine for Infectious Bronchitis

**DOI:** 10.3390/ani14121784

**Published:** 2024-06-13

**Authors:** Mengmeng Wang, Zongyi Bo, Chengcheng Zhang, Mengjiao Guo, Yantao Wu, Xiaorong Zhang

**Affiliations:** 1Jiangsu Co-Innovation Center for the Prevention and Control of Important Animal Infectious Disease and Zoonoses, College of Veterinary Medicine, Yangzhou University, Yangzhou 225009, China; dx120200166@stu.yzu.edu.cn (M.W.); zybo@yzu.edu.cn (Z.B.); zcc@yzu.edu.cn (C.Z.); guomj@yzu.edu.cn (M.G.); 2Joint International Research Laboratory of Agriculture and Agri-Product Safety, The Ministry of Education of China, Yangzhou University, Yangzhou 225009, China

**Keywords:** IBV, attenuated vaccine, genomic analysis, bioinformatics, mutation, virulence genes or sites

## Abstract

**Simple Summary:**

The live attenuated infectious bronchitis virus vaccine strain QXL87 was derived from the parental virulent strain JS/2010/12 through serial passages in specific pathogen-free (SPF) chicken embryos. However, the molecular mechanism underlying its attenuation remains unknown. In this study, the complete genome sequences of both strains were obtained through next-generation sequencing technology. By comparing amino acids, protein structures, and functional sites, three potential virulence genes (Nsp2, Nsp3, and S) were identified as likely contributors to altering viral pathogenicity. Furthermore, analysis of the effects of single amino acid mutations on protein function and stability revealed three potential functional mutation sites: P106S (Nsp2), A352T (Nsp2), and L472F (Nsp2). This research provides an initial insight into the molecular basis of IBV attenuation and is expected to facilitate the development of new vaccines.

**Abstract:**

The QXL87 live attenuated vaccine strain for infectious bronchitis represents the first approved QX type (GI-19 lineage) vaccine in China. This strain was derived from the parental strain CK/CH/JS/2010/12 through continuous passage in SPF chicken embryos. To elucidate the molecular mechanism behind its attenuation, whole-genome sequencing was conducted on both the parental and attenuated strains. Analysis revealed 145 nucleotide mutations in the attenuated strain, leading to 48 amino acid mutations in various proteins, including Nsp2 (26), Nsp3 (14), Nsp4 (1), S (4), 3a (1), E (1), and N (1). Additionally, a frameshift mutation caused by a single base insertion in the ORFX resulted in a six-amino-acid extension. Subsequent comparison of post-translational modification sites, protein structure, and protein–protein binding sites between the parental and attenuated strains identified three potential virulence genes: Nsp2, Nsp3, and S. The amino acid mutations in these proteins not only altered their conformation but also affected the distribution of post-translational modification sites and protein–protein interaction sites. Furthermore, three potential functional mutation sites—P106S, A352T, and L472F, all located in the Nsp2 protein—were identified through PROVEAN, PolyPhen, and I-Mutant. Overall, our findings suggest that Nsp2, Nsp3, and S proteins may play a role in modulating IBV pathogenicity, with a particular focus on the significance of the Nsp2 protein. This study contributes to our understanding of the molecular mechanisms underlying IBV attenuation and holds promise for the development of safer live attenuated IBV vaccines using reverse genetic approaches.

## 1. Introduction

Infectious bronchitis (IB) is an acute and highly contagious disease caused by the infectious bronchitis virus (IBV), affecting the respiratory, reproductive, nervous, and urinary systems of chickens [1,2,3]. In the global poultry industry, IBV is one of the most important viruses that seriously threatens the development of the industry [4,5]. A member of the Gammacoronavirus family, IBV is an unsegmented, single-stranded positive-sense RNA with a length of approximately 27.6 kb. The 3′ end of the genome encodes structural and accessory proteins, including the spike (S), accessory proteins 3a and 3b, envelope (E), membrane (M), 5a, 5b, and nucleocapsid (N) proteins. The 5′ end contains two overlapping open reading frames, ORF1a and ORF1b, which together constitute about two-thirds of the genome. ORF1a encodes polyprotein pp1a, while pp1ab is encoded by both ORF1a and ORF1b. These polyproteins are subsequently cleaved by virus-encoded enzymes, the papain-like proteinase (PLpro), and the 3C-like proteinase (3CLpro), into 15 nonstructural proteins (Nsp2–Nsp16) that play crucial roles in viral replication, transcription, host infection, and immune evasion [6,7,8,9,10,11].

To protect chickens against IBV infection, both inactivated and live attenuated vaccines are frequently used. Live attenuated vaccines, however, are preferred due to their higher immunogenicity, capable of inducing both humoral and cellular immune responses. These vaccines are typically obtained by serially passaging parental strains in SPF chicken embryos until the desired non-pathogenicity and immunogenicity are achieved [12]. However, this process is time-consuming, which is a significant limitation in the development of vaccines against emerging mutant strains. In addition, the use of live attenuated vaccines is constantly challenged by rapid viral variation, limited or poor cross-protection, and reversion to virulence [13]. A clearer analysis and targeted modification of the genes and sites associated with IBV virulence could significantly enhance the development and effectiveness of IBV vaccines.

Previous studies have highlighted that among the IBV structural proteins, the S protein experiences the most mutations during the attenuation process, indicating its key role in pathogenicity [14,15]. However, the pathogenicity of the Beau-R attenuated strain remained unchanged when expressing the S protein of the M41 pathogenic strain, suggesting that the S protein is not the sole contributor to IBV pathogenicity [16]. Conversely, the pathogenic M41 strain expressing the replicase gene of the attenuated Beaudette strain lost its pathogenicity, indicating that the replicase protein may be a determinant of IBV pathogenicity [17]. Genome sequence alignment between parental and attenuated IBV strains revealed multiple mutations in the replicase gene ORF1a, and the reverse genetic system was used to construct recombinant viruses. It was found that the ORF1a gene of IBV is a determinant of pathogenicity [18]. Additionally, studies have shown that deletions of 3a, 3b, 5a, and 5b result in mutant viruses with an attenuated phenotype, suggesting that accessory proteins are also involved in the regulation of viral pathogenicity [19,20].

The QXL87 live attenuated vaccine strain for infectious bronchitis, which belongs to the GI-19 lineage [21], has been officially approved as the first vaccine for this lineage in China. This strain was derived from the parental strain CK/CH/JS/2010/12 by continuous passage in SPF chicken embryos. To elucidate the changes in genetic characteristics associated with IBV attenuation, the whole genome of the parental strain JS/2010/12 and the attenuated strain QXL87 were sequenced and analyzed in this study. Amino acids, structures, and functional sites in viral proteins from both strains were compared, and three genes (Nsp2, Nsp3, and S) with notable differences were identified. Furthermore, through analysis of the impact of amino acid mutations on protein structure and stability, three functional mutation sites (P106S, A352T, and L472F) were identified. These insights into potential virulence genes and sites are crucial for the development of new, more effective IBV vaccines.

## 2. Materials and Methods

### 2.1. Viruses

The virulent JS/2010/12 strain was isolated by our laboratory in 2010 from chickens with respiratory and renal disease. After culturing the strain in SPF chicken embryos for 5 generations, the parental strain JS/2010/12 (GenBank accession number PP100175), which can stably replicate in chicken embryos, was obtained. The parental strain caused the partial death of SPF chickens, with autopsy revealing inflammatory or caseous exudate in the trachea and bronchus, enlarged kidneys with urate deposits, and signs of inflammation or dysplasia in the ovaries [22,23]. The attenuated vaccine strain QXL87 (GenBank accession number PP100176) was derived from the parental strain JS/2010/12 through 120 serial passages in SPF chicken embryos. No clinical response was observed when QXL87 was inoculated. Meanwhile, no microscopic changes were detected in tissue.

### 2.2. Next-Generation Sequencing (NGS)

The parental JS/2010/12 strain and the attenuated vaccine QXL87 strain were separately inoculated into the allantoic cavity of 9-day-old SPF chicken embryos. After propagation at 37 °C for 48 h, the allantoic fluid was collected, and the cellular debris was removed by centrifugation at 5000× *g* for 10 min. The supernatant was then enriched using a 100K Centrifugal Ultrafiltration Device (Pall Corporation, New York, NY, USA). RNA was extracted from the supernatant using an Ultrapure RNA Kit (CWBIO, Taizhou, China) following the manufacturer’s instructions. Subsequently, a double-stranded cDNA library ranging from 250 to 500 bp was prepared using the TruSeq TM DNA Sample Prep Kit (Illumina, San Diego, CA, USA). The constructed library was subjected to paired-end (PE) sequencing on the NGS Illumina HiSeq2500 platform. The raw data were processed using Fastp v 0.20.0 (https://github.com/OpenGene/fastp, accessed on 7 March 2023) to eliminate sequencing adapters and low-quality reads with a quality score below Q20. Ribosomal RNAs and host read subtraction by read-mapping were accomplished with BBMAP (https://github.com/BiolnfoTools/BBMap, accessed on 8 March 2023). De novo genome assembly was carried out using SPAdes v3.14.1 (https://github.com/ablab/spades, accessed on 9 March 2023) to achieve the complete genome sequence. Finally, the assembled reads were compared using the NCBI BLAST database. This experiment was performed with the assistance of Tanpu Biotechnology Co., Ltd. (Shanghai, China).

### 2.3. Prediction of Protein Post-Translational Modification Sites

The post-translational modifications (PTMs) sites of Nsp2, Nsp3, Nsp4, S, 3a, E, N and ORFX proteins from both parental and attenuated strains were individually predicted to assess the impact of amino acid mutations on their modifications. The online analysis tools employed were as follows: NetPhos 3.1 (https://services.healthtech.dtu.dk/services/NetPhos-3.1, accessed on 10 March 2024) for phosphorylation site prediction, with a prediction score above 0.500 considered positive; NetNGlyc (https://services.healthtech.dtu.dk/services/NetNGlyc-1.0/, accessed on 13 March 2024) and NetOGlyc (https://services.healthtech.dtu.dk/services/NetOGlyc-4.0/, accessed on 22 March 2024) for glycosylation site prediction; and CUCKOO (http://biocuckoo.cn, accessed on 17 April 2024) to predict protein methylation, lysine acetylation, and ubiquitination sites.

### 2.4. Homology Modeling

The automatic mode of SWISS-MODEL (https://swissmodel.expasy.org/, accessed on 30 December 2023) was employed to identify homologous sequences of the target proteins. Proteins displaying high amino acid similarity and extensive coverage were chosen as templates for homology modeling. The three-dimensional structure of the protein was visualized using the PyMOL 3.0 system. To evaluate the structural changes caused by non-synonymous substitutions, the root mean square deviation (RMSD) of relevant proteins between parental and attenuated strains was calculated.

### 2.5. Prediction of Protein-Protein Binding Sites

The protein–protein binding sites of the parental and attenuated strains Nsp2, Nsp3, Nsp4, S, 3a, E, and N proteins were predicted through the ScanNet online website (http://bioinfo3d.cs.tau.ac.il/ScanNet/, accessed on 20 April 2024). Following this, the influence of amino acid mutations on protein–protein binding affinity was evaluated. ScanNet (Spatio-Chemical Arrangement of Neighbors Network), a geometric deep learning model, facilitates structure-based prediction of binding sites. By inputting the raw structural file, ScanNet iteratively builds representations of atoms and amino acids based on the spatio-chemical arrangement of their neighbors. These representations are then used to predict the probability of amino acid involvement in binding sites [24,25].

### 2.6. Prediction of Potential Virulence Sites

In this study, the functional prediction of amino acid mutation sites was performed using PROVEAN (http://provean.jcvi.org/seq_submit.php, accessed on 13 April 2024) and Polyphen (http://genetics.bwh.harvard.edu/pph2/index.shtml, accessed on 13 April 2024). PROVEAN is designed to predict the effects of single or multiple amino acid substitutions, insertions, and deletions. Variants with a score of −2.5 or below are considered “deleterious” and those scoring above −2.5 are considered “neutral” mutations. Polyphen-2 utilizes a machine-learning approach based on the naïve Bayes model to predict the likelihood of amino acid mutations affecting protein structure and function. A threshold of 0.50 is used, with scores closer to 1.0 indicating a higher probability of influencing protein function. Additionally, I-Mutant (http://folding.biofold.org/i-mutant/i-mutant2.0.html, accessed on 16 April 2024) was used to predict the impact of amino acid mutations on protein stability. DDG < 0 indicates decreased protein stability, and DDG > 0 indicates increased protein stability.

## 3. Results

### 3.1. NGS Analysis

To investigate the changes during IBV attenuation, total RNA was extracted from both the parental JS/2010/12 strain and the attenuated vaccine QXL87 strain for NGS analysis. The parental strain generated 338,276 reads, while the attenuated strain generated 797,639 reads, with average sequencing depths of 6771.25× and 8058.06×, respectively (Figure 1A,B). Excluding the polyA sequence, the genome sizes of the parental and attenuated strains were 27,664 nt and 27,665 nt, respectively. Both genomes consisted of 5′ UTR, ORF1a, ORF1b, S, 3a, 3b, E, M, ORFX, 5a, 5b, N, and 3′ UTR. The sequence identity between the two strains reached 99.47%. In the attenuated strain, a total of 145 base substitutions were identified across various regions, including 5′ UTR (4), Nsp2 (80), Nsp3 (50), Nsp4 (1), Nsp9 (1), Nsp14 (1), S (5), 3a (1), E (1), and N (1). The mutation rates for the mentioned genes were as follows: 0.76%, 4.00%, 1.05%, 0.06%, 0.30%, 0.06%, 0.14%, 0.57%, 0.31%, and 0.08% (Figure 1C). Furthermore, a nucleotide insertion was observed in the ORFX region. Amino acid sequence alignment showed that 53 non-synonymous mutations resulted in 48 amino acid residue changes, distributed across Nsp2 (26), Nsp3 (14), Nsp4 (1), S (4), 3a (1), E (1), and N (1). Additionally, a frameshift mutation in the ORFX protein resulted in the addition of six amino acids. The distribution of amino acid differences between the parental and attenuated strains is shown in Figure 1D. Notably, the Nsp2 gene had the highest number and rate of nucleotide mutations, consequently causing the most significant amino acid residue alterations.

### 3.2. Difference Analysis of PTMs

PTMs are crucial for regulating the replication, infection, and pathogenicity of coronaviruses [26]. This study analyzed mutations in phosphorylation, methylation, lysine acetylation, N-glycosylation, O-glycosylation, and ubiquitination sites across viral proteins from both parental and attenuated strains. The analysis revealed PTM mutations in Nsp2, Nsp3, S, E, and ORFX proteins following IBV attenuation. In the attenuated strain, the phosphorylation sites 191T, 328S, and 447T of Nsp2 were absent, while new phosphorylation sites were discovered at positions 105S, 106S, 224S, 352T, 562S, and 611T. The protein kinases associated with these phosphorylation site mutations included PKA (224S), PKC (191T), PKG (447T), p38MAPK (352T), DNAPK (562S), CKI (328S), and CKII (328S, 611T). Meanwhile, a new lysine acetylation site (110K) and an N-glycosylation site (178N) were identified in the attenuated strain (Figure 2A). Compared with the parental strain, the attenuated strain Nsp3 exhibited an increase in phosphorylation sites at 235P and 1570T, along with a new lysine acetylation site at 206K, while five O-glycosylation sites (122T, 132T, 185S, 189T, and 518S) were absent (Figure 2B). The protein kinases responsible for the phosphorylation site mutations in Nsp3 were PKC (235S) and CDC2 (1570T). The S protein introduced two new phosphorylation sites, 67T and 120S, while losing the 548S phosphorylation site (Figure 2C). Protein kinases PKC (67T, 120S) and CDC2 (548S) were implicated in these phosphorylation site mutations. Additionally, the E protein lost one lysine acetylation site 71K (Figure 2D) and the ORFX protein lost two lysine acetylation sites 92K and 93K (Figure 2E).

### 3.3. Difference Analysis of Protein Three-Dimensional Structure

Considering the amino acid mutations in Nsp2, Nsp3, Nsp4, S, 3a, E, and N or additions in ORFX proteins during IBV attenuation, we further investigated their effect on protein structure. The qualifying homology models were selected as templates and the following regions of each protein were modeled: Nsp2 (14–597 aa), Nsp3 (337–501 aa, 504–804 aa), Nsp4 (419–513aa), S (21–1025 aa), 3a (9–46 aa), E (10–65 aa), and N (18–362 aa). However, the structural model of ORFX could not be constructed due to the lack of a suitable template in the database. By calculating the RMSD between parental and attenuated strains, we found that amino acid mutations influenced the spatial conformation of Nsp2 (0.027), Nsp3 (0.008/0.000), Nsp4 (0.005), S (0.002), and N (0.002) proteins (Figure 3A–E). The Nsp2 protein structure showed the most significant alterations due to amino acid mutations. This observation was further supported by an overlap analysis of protein monomer structures. Following IBV attenuation, a prominent change was detected at amino acid 565 of the Nsp2 protein, which changed from a helical to a coiled structure (Figure 3A). To better characterize the structural implications of amino acid mutations, these mutations were localized on its three-dimensional structural model (Figure 3F–J).

### 3.4. Difference Analysis of Protein–Protein Binding Sites

Protein–protein interactions play a crucial role in various cellular processes, including signal transduction, cell growth, proliferation, and apoptosis. These interactions are mainly determined by their binding affinities. Studying how amino acid substitutions affect the binding capabilities of proteins with their interaction partners can provide valuable insights into viral pathogenesis [27]. In this study, the ScanNet server was utilized to perform a comparative analysis of protein–protein binding sites in parental and attenuated strains of Nsp2, Nsp3, Nsp4, S, 3a, E, and N based on protein structures. The findings revealed significant alterations in the protein–protein binding sites of Nsp2, Nsp3, and S following IBV attenuation. Specifically, the protein binding affinity of amino acids 68, 104, 109, and 496–597 in the Nsp2 protein (Figure 4A), as well as amino acid 548 in the S protein (Figure 4C), exhibited a notable increase. Conversely, the binding affinity of amino acids 460, 466, and 467 in the Nsp3 protein decreased (Figure 4B). Further details on the comparison of protein–protein binding sites for other proteins can be found in Appendix A.

### 3.5. Prediction and Analysis of Virulence Sites on IBV Genome

Amino acid mutations might impact DNA transcription and translation, leading to changes in protein structure and function [28]. The effects of amino acid mutations in the viral genome on protein function were predicted using PROVEAN and PolyPhen software. PROVEAN identified six deleterious mutation sites, while PolyPhen identified ten damaging mutation sites. I-Mutant software was used to evaluate the impact of amino acid mutations on protein stability. Out of 48 mutation sites analyzed, 37 were found to decrease protein stability (Table 1). A Venn diagram analysis pinpointed three significant functional mutation sites: P106S (Nsp2), A352T (Nsp2), and L472F (Nsp2) (Figure 5A). Three-dimensional protein structure comparison showed that all three functional mutation sites were involved in the structural changes in the Nsp2 protein, where the 565th amino acid changed from a helical to a coiled structure (Figure 5B–D). Further analysis revealed that the P106S mutation introduced new protein phosphorylation (105S and 106S) and lysine acetylation sites (110 K) (Figure 5E) and significantly enhanced the protein binding affinities at amino acid positions 68, 104, 106, and 109 (Figure 5G). The A352T mutation introduced a new phosphorylation site (352T) and reduced the protein binding affinity at amino acid 470 (Figure 5F,H). The L472F mutation enhanced the protein binding affinity at amino acids 569, 571, and 572 (Figure 5I).

## 4. Discussion

Serial passage of parental IBV in SPF chicken embryos is a common method for developing attenuated vaccines [29]. In our laboratory, this method was successfully utilized to obtain the live attenuated vaccine strain QXL87 for the QX type, which has since become one of the most widely used live vaccines in China. However, the molecular mechanism behind its attenuation remains unclear. In this study, the genome sequences of the parental strain JS/2010/12 and its attenuated strain QXL87 were compared and analyzed. The results showed that mutations were distributed across the entire genome, affecting Nsp2, Nsp3, Nsp4, S, 3a, E, N, and ORFX proteins. Subsequently, we compared the sequence changes in our results to those reported in other studies and found no universal differences in the amino acid mutations responsible for IBV attenuation [30,31]. This implies that the attenuation of IBV may be caused by multiple amino acid mutations.

In this study, it was found that 90.34% of all amino acid substitutions in the entire genome were located in the ORF1a gene, indicating the crucial role of ORF1a in IBV attenuation. Similar findings were observed in a comparative analysis of pathogenic and attenuated vaccine strains of Ark DPI strains [15]. Moreover, the key role of the YN strain ORF1a gene in IBV pathogenicity has been determined based on a reverse genetic system [18]. By analyzing the position and number of amino acid mutations in ORF1a, it was discovered that there were 26, 14, and 1 amino acid substitutions in the Nsp2, Nsp3, and Nsp4 proteins, respectively, between the parental strain and the attenuated strain. Previous studies have also reported amino acid substitutions in these proteins [32,33]. One study pointed out that the Nsp3 protein showed the most amino acid differences during the attenuation of Ark, GA98, and Mass41 strains [34]. However, in our study, although Nsp3 had a large number of amino acid mutations, Nsp2 was found to exhibit the most amino acid differences during viral attenuation. Importantly, this phenomenon was reported for the first time in our related study. Besides ORF1a, amino acid mutations were also identified in the S, 3a, E, N, and ORFX proteins in this study. Previous research has shown the involvement of S, 3a, E, and N proteins as virulence genes in regulating IBV pathogenicity [19,20,35,36]. However, the role of ORFX in viral pathogenicity remains unclear. Not all IBVs contain this ORF [37], and its generation of mutations during IBV virulence was reported for the first time in our study. We propose that this protein may play a role in regulating viral pathogenicity, but further experiments are needed to confirm this hypothesis.

PTMs alter amino acid side chains without changing amino acid sequences, serving as subtle mechanisms for maintaining cellular protein homeostasis and enabling complex functions. Common PTMs include phosphorylation, glycosylation, methylation, acetylation, and ubiquitination. During coronavirus–host interactions, viruses promote viral proliferation and immune evasion through PTMs [38]. In this study, phosphorylation site mutations were observed in Nsp2, Nsp3, and S proteins after IBV attenuation. Given the critical role of phosphorylation in coronavirus replication, transcription, and assembly [39,40,41], we hypothesize that the introduction of phosphorylation sites in these proteins could enhance viral proliferation. The N-glycosylation of the coronavirus Nsp4 protein plays a crucial role in the formation of double-membrane vesicles (DMVs) [42]. In this study, a new N-glycosylation site was introduced in the Nsp2 protein, which is known to be recruited to the DMV and convoluted membrane (CM) during viral replication. We speculate that the introduction of this site could enhance viral replication. Previous research has shown that recombinant mouse hepatitis virus (MHV) containing O-glycosylated M protein induces higher levels of type I interferon compared to MHV with non-glycosylated M protein [43]. Following IBV attenuation, our study uncovered a deletion of an O-glycosylation site within the PLPro structural domain of the Nsp3 protein, which has been shown to have interferon antagonistic functions [44]. We speculate that the deletion of O-glycosylation could aid the virus in escaping the innate immune response, thereby promoting viral infection. In addition, this study identified acetylation site mutations in Nsp2, Nsp3, E, and ORFX proteins. Previous research has shown that pp1ab of MERS-CoV employs the host acetylation mechanism to regulate enzymatic activity for optimal replication [45]. Therefore, we hypothesized that acetylation sites introduced in the Nsp2 and Nsp3 proteins may enhance viral replication. Acetylation of K53 and K63 at the C-terminus of the SARS-CoV-2 E protein enhanced its interaction with BRD4, thereby impacting the host immune response [46]. In our study, the acetylation sites at K53 and K63 were not found on the C-terminus of the IBV E protein, while an acetylation site at K71 was identified. Interestingly, this site was lost during IBV attenuation. We speculate that this mutation may impact the regulation of immune responses. Due to the lack of studies on the ORFX protein, we are unable to ascertain the effect of mutations in the acetylation site of this protein on the regulation of viral pathogenicity.

Alterations in the structure of a protein can often impact its function [47]. In this study, the spatial conformations of Nsp2, Nsp3, Nsp4, S and N proteins were altered, suggesting a potential relationship between these proteins and IBV pathogenicity. Subsequently, we further analyzed the effect of amino acid mutations on protein–protein binding sites based on the three-dimensional structure of the proteins. The protein binding affinity of Nsp2 protein 496–597aa was found to be significantly enhanced, possibly due to the alteration of amino acid 565 from a helical to a random coil structure. Previous studies have demonstrated that Nsp2 proteins interact with various host proteins and play a role in regulating viral replication, transcription, and immune response [48,49,50]. We supposed that changes in Nsp2 protein structure and protein binding affinity are important regulators of viral pathogenicity. The altered the protein binding affinity of the Nsp3 protein occurring within the macrostructural domain. This domain can influence viral pathogenesis and immune regulation by interacting with other proteins [51]. Reduced protein binding affinity in this domain may alter protein interactions and thereby affect the regulation of viral pathogenicity by Nsp3 proteins. Additionally, our study identified a notable increase in protein binding affinity at amino acid 548 of the S protein, attributed to the S548R mutation. This mutation could potentially influence the spatial conformation of the protein through phosphorylation modifications, thereby changing its ability to interact with other molecules. The host proteins found to interact with S proteins are primarily related to components of the innate immune pathway, such as cytokines and chemokines [52]. We speculated that changes in the interaction site of the IBV S protein could impact its function in the regulation of innate immunity.

In this study, the amino acid mutations occurring in Nsp2, Nsp3, and S affected not only the spatial conformation of the proteins but also the distribution of post-translational modification sites and protein–protein binding sites. We hypothesized that they are important genes involved in the regulation of viral pathogenicity. The involvement of these three proteins as virulence factors in the regulation of pathogenicity in other coronaviruses has been demonstrated [53,54]. However, the role of the other two proteins, except for the S protein [20], in the regulation of IBV pathogenicity needs to be further verified by reverse genetic systems.

Amino acid mutations have the potential to impact protein function and consequently viral pathogenicity. Computer simulation methods provide an effective means of identifying deleterious mutations in specific genes [55]. In this study, functional amino acid mutation site prediction was performed using three software packages: PROVEAN, PolyPhen, and I-Mutant. Since these methods are both independent and complementary to each other, a comprehensive evaluation of these prediction methods was conducted to minimize discrepancies in the results. Three functional mutation sites were finally screened, which were P106S (Nsp2), A352T (Nsp2), and L472F (Nsp2). Notably, all three mutations were found in Nsp2. PCR amplification and Sanger sequencing were conducted on the Nsp2 genes of JS/2010/12 and QXL87 strains to reconfirm the presence of these important mutation sites. Further analysis indicated that these mutations could alter protein structure and biological functions by affecting post-translational modifications or protein–protein interactions. This finding further supports the hypothesis that Nsp2 plays a critical role as a virulence factor in regulating IBV pathogenicity. Despite providing insights into the damaging effects of amino acid mutations, additional biological validation through investigating protein function or creating mutant viruses using reverse genetic systems is essential to comprehend their specific roles and impact on viral pathogenicity.

## 5. Conclusions

In conclusion, this study obtained the whole-genome sequences of the parental JS/2010/12 strain and the attenuated vaccine QXL87 strain and identified several amino acid mutations in the proteins. The amino acid mutations in Nsp2, Nsp3, and S proteins not only altered the spatial conformation of the proteins but also affected the distribution of post-translational modification sites and protein–protein interaction sites. Therefore, it was hypothesized that Nsp2, Nsp3, and S proteins play crucial roles as virulence genes in altering viral pathogenicity. Furthermore, three potential virulence sites, P106S (Nsp2), A352T (Nsp2), and L472F (Nsp2), were identified by bioinformatics software. These findings provide a theoretical basis for investigating the molecular mechanisms of IBV attenuation and offer valuable insights for the development of innovative IBV vaccines.

## Figures and Tables

**Figure 1 animals-14-01784-f001:**
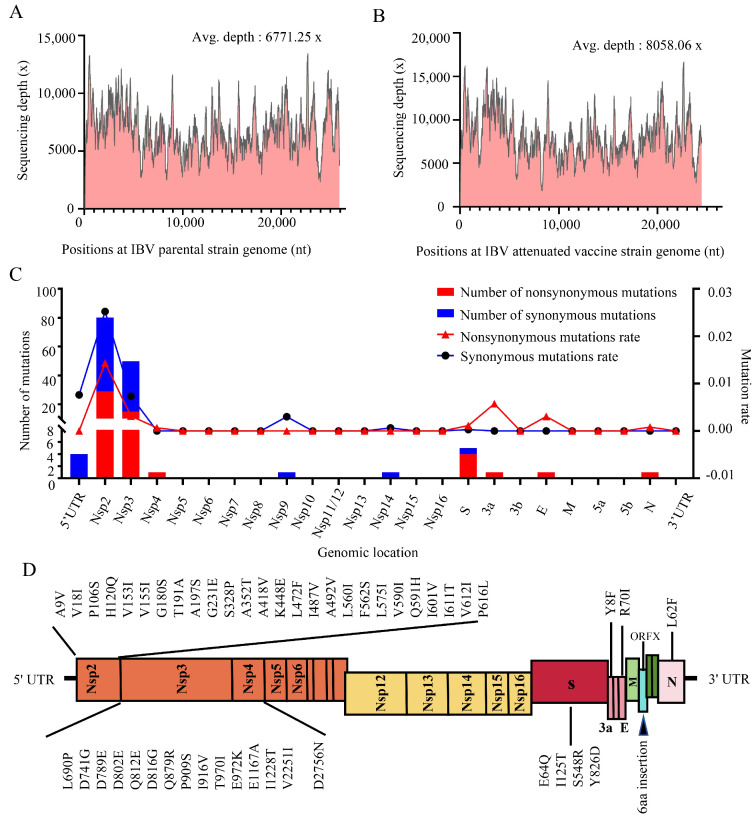
Next-generation sequencing analysis of the parental and attenuated IBV strains. (**A**) Sequencing depth for the IBV parental strain. (**B**) Sequencing depth for the attenuated IBV strain. (**C**) Per genomic location with number and mutation rate of nonsynonymous and synonymous mutations. The bar graph illustrates the number of mutations at each genomic location, with blue bars representing synonymous mutations and red bars representing non-synonymous mutations. Meanwhile, the line graph shows the mutation rates for genes, with blue lines indicating synonymous mutation rates and red lines indicating non-synonymous mutation rates. (**D**) Distribution of amino acid mutations across the parental and attenuated strains of IBV.

**Figure 2 animals-14-01784-f002:**
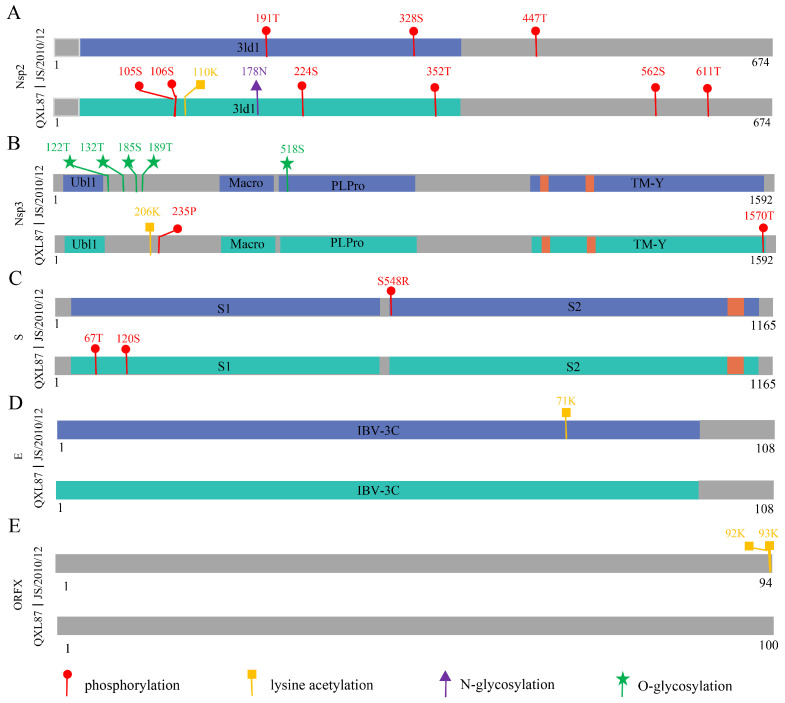
Schematic diagram of PTM mutations of Nsp2 (**A**), Nsp3 (**B**), S (**C**), E (**D**), and ORFX (**E**) proteins during IBV attenuation.

**Figure 3 animals-14-01784-f003:**
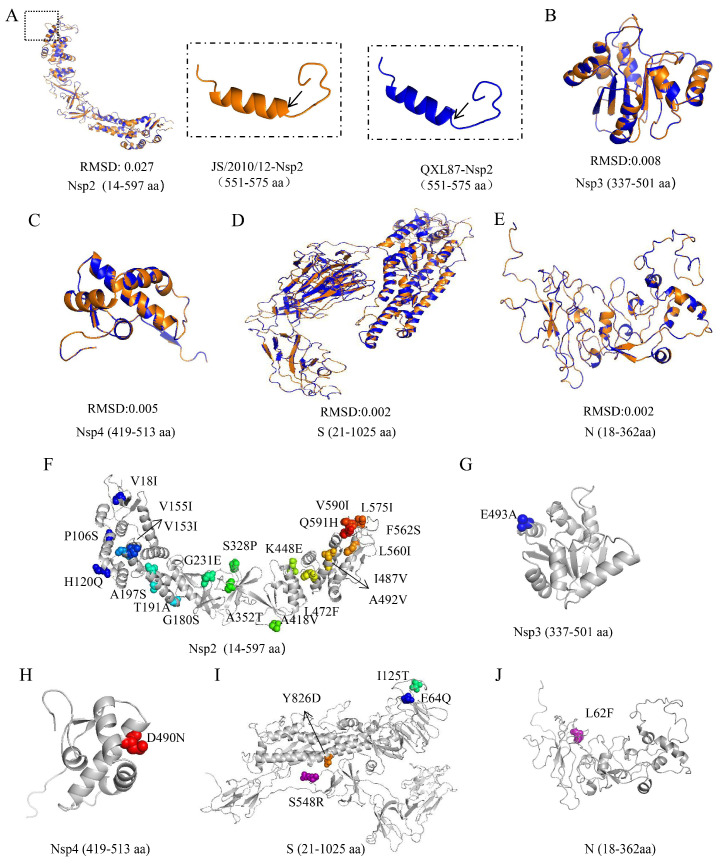
Three-dimensional structural analysis of proteins that changed between parental and attenuated strains. The monomer structural overlap of Nsp2 (**A**), Nsp3 (**B**), Nsp4 (**C**), S (**D**), and N (**E**) proteins between parental and attenuated strains. Cartoon model of Nsp2 (**F**), Nsp3 (**G**), Nsp4 (**H**), S (**I**), and N (**J**) proteins where the mutated amino acids are shown.

**Figure 4 animals-14-01784-f004:**
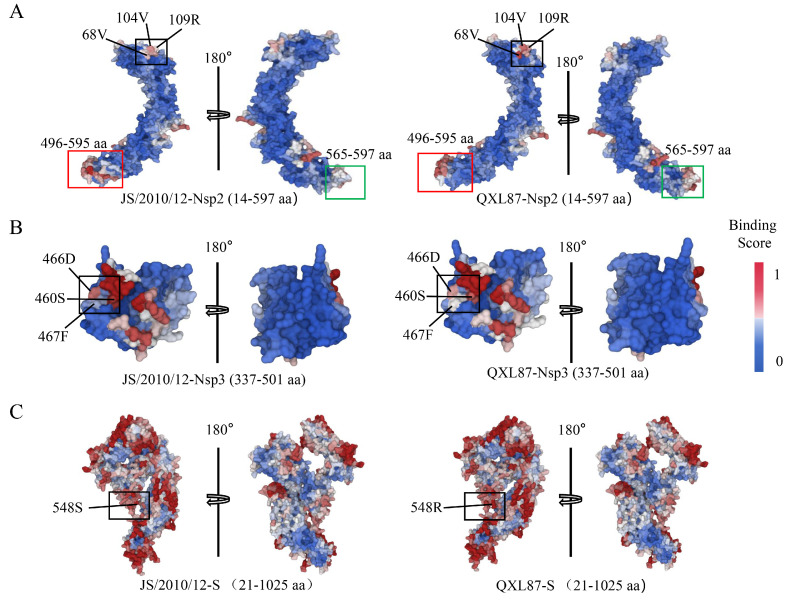
Prediction and comparison results of protein binding sites in Nsp2 (**A**), Nsp3 (**B**), and S (**C**) proteins of parental and attenuated strains, with colors indicating binding propensity, from low (blue) to high (red).

**Figure 5 animals-14-01784-f005:**
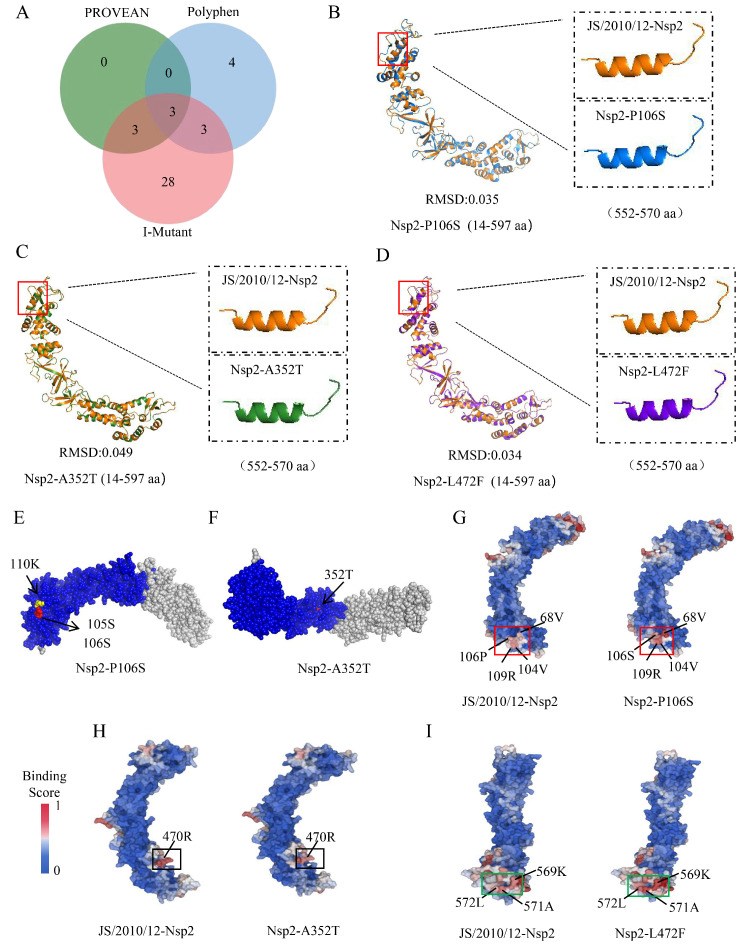
Prediction and analysis of functional amino acid mutation sites on IBV genome. (**A**) Venn diagrams of functional amino acid mutation site prediction in IBV genome. Prediction of functional effects of amino acid mutation sites on protein three-dimensional structure (**B**–**D**), PTMs (**E**,**F**), and protein–protein binding sites (**G**–**I**).

**Table 1 animals-14-01784-t001:** Prediction of functional amino acid mutation sites on IBV genome.

Protein Name	Mutation Site	PROVEAN	PolyPhen	I-Mutant (DDG)
Nsp2	A9V	Neutral	Neutral	−0.9 KJ/mol
	V18I	Neutral	Neutral	−0.39 KJ/mol
	P106S	Deleterious	Damaging	−1.42 KJ/mol
	H120Q	Neutral	Neutral	−0.37 KJ/mol
	V153I	Neutral	Neutral	−0.16 KJ/mol
	V155I	Neutral	Neutral	−0.19 KJ/mol
	G180S	Neutral	Neutral	−1.33 KJ/mol
	T191A	Neutral	Neutral	−1.31 KJ/mol
	A197S	Deleterious	Neutral	−0.29 KJ/mol
	G231E	Neutral	Damaging	−0.09 KJ/mol
	S328P	Neutral	Neutral	−0.91 KJ/mol
	A352T	Deleterious	Damaging	−1.24 KJ/mol
	A418V	Neutral	Damaging	0.11 KJ/mol
	K448E	Neutral	Neutral	0.49 KJ/mol
	L472F	Deleterious	Damaging	−0.24 KJ/mol
	I487V	Neutral	Neutral	−0.95 KJ/mol
	A492V	Neutral	Neutral	−0.41 KJ/mol
	L560I	Neutral	Neutral	0.41 KJ/mol
	F562S	Neutral	Neutral	−1.35 KJ/mol
	L575I	Neutral	Neutral	−0.51 KJ/mol
	V590I	Neutral	Neutral	−1.24 KJ/mol
	Q591H	Deleterious	Neutral	−0.92 KJ/mol
	I601V	Neutral	Neutral	−1.93 KJ/mol
	I611T	Neutral	Neutral	−2.39 KJ/mol
	V612I	Neutral	Neutral	0.27 KJ/mol
	P616L	Neutral	Neutral	−0.16 KJ/mol
Nsp3	L16P	Neutral	Neutral	−1.01 KJ/mol
	D67G	Neutral	Damaging	0.23 KJ/mol
	D115E	Neutral	Neutral	−0.2 KJ/mol
	D128E	Neutral	Neutral	0.76 KJ/mol
	Q138E	Neutral	Neutral	0.55 KJ/mol
	D142G	Neutral	Neutral	−0.01 KJ/mol
	Q205R	Neutral	Neutral	−0.73 KJ/mol
	P235S	Neutral	Neutral	−0.29 KJ/mol
	I242V	Neutral	Neutral	−1.18 KJ/mol
	T296I	Neutral	Neutral	0.82KJ/mol
	E298K	Neutral	Neutral	−0.22 KJ/mol
	E493A	Neutral	Neutral	−0.9 KJ/mol
	I554T	Neutral	Neutral	−2.22 KJ/mol
	V1577I	Neutral	Neutral	−0.31 KJ/mol
Nsp4	D490N	Neutral	Damaging	−2.88 KJ/mol
S	E64Q	Neutral	Neutral	−0.48 KJ/mol
	I125T	Neutral	Neutral	−1.08 KJ/mol
	S548R	Neutral	Damaging	0.78 KJ/mol
	Y826D	Neutral	Neutral	−1.35 KJ/mol
3a	Y8F	Neutral	Neutral	0.78 KJ/mol
E	R70I	Neutral	Damaging	−0.52 KJ/mol
N	L62F	Neutral	Neutral	1.11 KJ/mol

## Data Availability

Currently, the data are not publicly available. However, the sequence data presented in this study will become publicly accessible on NCBI starting in June 2025.

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
