# Peer review of "Deciphering the Genetic Variation: A Comparative Analysis of Parental and Attenuated Strains of the QXL87 Vaccine for Infectious Bronchitis"

_animals, 2024, doi:10.3390/ani14121784_

Round 1

Reviewer 1 Report

Comments and Suggestions for Authors

Coronaviruses have long been an production problem of poultry and other animal protein systems. Producers have been attempting to insulate themselves from disease related losses using MLVs for decades with mixed results. Attempting to understand the process of attenuation has been unsuccessful to date in part due to the lack of uncomplicated, reverse engineering model for coronaviruses such as IBV.  The authors have applied modern sequencing approaches and bioinformatics tools to attempt to explain the process of attenuation through the recognition of sequence signatures resulting in the phenotypic changes, mainly attenuation. The manuscript was well written and I find no issue with the approach. I offer the following comments for the authors to consider.

1.  Line 95: The parental JS/2010/12 strain and the attenuated vaccine QXL87 strain were separately inoculated into the allantoic cavity of 9-day-old SPF chicken embryos. 

 There is no mention in the manuscript of the potential impact of this step on the composition of the viral population that ultimately is sequenced. This may influence the population with the JS/1010/12 strain. Also, what is the embryo passages were performed when creating the stock of JS/2010/12 used in this study? Again, a high number of embryo passages could influence the population of viruses sequenced. Some have suggested that even 1 embryo passage will selected for a population of viruses that is different from those that replicate within the bird. 

2. I do not believe that I saw how SNPs were resolved when creating a consensus sequence. Do the programs utilized in this study take into account all the SNPs recognized at a given location? Is a consensus sequence created for the JS/2010/12 strain and then the vaccine strain is compared to that sequence as a reference guided assembly? Since the original assembly process is very important, I would like to see additional details described. 

3. Do the subsequent bioinformatics tools utilize the consensus sequence containing all SNPs or do they handle all the raw reads? 

4. It does not appear that any of the significant SNPs were confirmed by Sanger sequencing. Is this correct? I did not see a mention of Illumina's error rate and the potential impact on the identification of SNPs.

5. There is no attempt to biologically prove the impact of the SNPs.  There is no mention of the need to do this. There is a great deal of suggesting, speculating and supposing in the discussion. Would be nice to see a proposal of how the SNPs may be validated.  

Reviewer 2 Report

Comments and Suggestions for Authors

The manuscript by Wang et al describes the results of whole-genome sequencing of two strains of infectious bronchitis virus (IBV). This is the same virus belonging to the QX lineage (GI-19), one strain JS/2010/12 sequenced shortly after its identification and isolation from diseased chickens in the field, and the other QXL87 is the same virus after multiple passages on SPF eggs prepared as a vaccine strain. The authors compared the two sequences, looking for whether the detected nucleotide changes are synonymous or non-synonymous and, if so, whether they affect subsequent post-transcriptional modifications such as phosphorylation, N-glycosylation, O-glycosylation, methylation, lysine acetylation or ubiquitination. Protein-protein binding sites and analysis of the functional effect of the detected nsSNPs were also assessed. These all-component bioinformatics analyses enabled the identification of three genes in which alterations may lead to attenuation. In these two strains, three aa sites in nsp2 were identified that may have an especially impact on attenuation.

Studies on the mechanisms of attenuation of IBV strains has been ongoing for years and the manuscript submitted for review brings a lot of interesting information. I consider the paper to be well written and documented and worthy of publication.

However I have also a few comments:

In 2016, a new nomenclature was proposed to describe the IBV strain type (Valastro et al, 2016). To my knowledge QX strains are the GI-19 lineage - please include this information.

I am missing information on the attenuation process of this field strain - how many passages did this attenuation require - 20, 30 or 100 passages? How was this strain assessed for attenuation? What changes/disease did the parent strain cause (was it respiratory or renal or digestive). I am missing this knowledge!!!

In addition, the use of attenuated vaccines causes other problems in the field - the necessity to differentiate during diagnostics whether we may be dealing with a field or a vaccine strain. Can any changes detected by the authors of this manuscript be used to distinguish vaccine from field strains?

I don't know of an IBV strain that affect the nervous system as stated in line 43 - this is probably a mistake.

I also found inaccuracies in Fig. 1 c - is the scale in this diagram correct?

Author Response

Response to concerns raised by Reviewer #2

The manuscript by Wang et al describes the results of whole-genome sequencing of two strains of infectious bronchitis virus (IBV). This is the same virus belonging to the QX lineage (GI-19), one strain JS/2010/12 sequenced shortly after its identification and isolation from diseased chickens in the field, and the other QXL87 is the same virus after multiple passages on SPF eggs prepared as a vaccine strain. The authors compared the two sequences, looking for whether the detected nucleotide changes are synonymous or non-synonymous and, if so, whether they affect subsequent post-transcriptional modifications such as phosphorylation, N-glycosylation, O-glycosylation, methylation, lysine acetylation or ubiquitination. Protein-protein binding sites and analysis of the functional effect of the detected nsSNPs were also assessed. These all-component bioinformatics analyses enabled the identification of three genes in which alterations may lead to attenuation. In these two strains, three aa sites in nsp2 were identified that may have an especially impact on attenuation.

Studies on the mechanisms of attenuation of IBV strains has been ongoing for years and the manuscript submitted for review brings a lot of interesting information. I consider the paper to be well written and documented and worthy of publication.

However I have also a few comments:

Comment #1: In 2016, a new nomenclature was proposed to describe the IBV strain type (Valastro et al, 2016). To my knowledge QX strains are the GI-19 lineage - please include this information.

Response: Thank you for your suggestion. Although we identified the QX strain as belonging to the GI-19 lineage in our article, we acknowledge the lack of citations to substantiate this classification. We have addressed this issue by providing an explanation and referencing relevant literature (Please see lines 80-82 and 468-470).

Comment #2: I am missing information on the attenuation process of this field strain - how many passages did this attenuation require - 20, 30 or 100 passages? How was this strain assessed for attenuation? What changes/disease did the parent strain cause (was it respiratory or renal or digestive). I am missing this knowledge!!!

Response: Thank you for your insightful comment. The attenuated vaccine strain QXL87 (GenBank accession No. PP100176) was derived from the parental strain JS/2010/12 after undergoing 120 consecutive passages in SPF chicken embryos. Previous experiments in our laboratory assessed the pathogenicity of different generations in chickens. Results indicated that the parental JS/2010/12 strain was lethal to SPF chickens, causing significant inflammatory or caseous exudate in the trachea and bronchus, enlarged kidneys with urate deposits in the ureter, giving them a marbled appearance, and inflammation or dysplasia of the ovaries upon autopsy [1-2]. In contrast, SPF chickens inoculated with the QXL87 strain showed no signs of illness or mortality, with all tissues and organs appearing normal. These details have been documented in the “Materials and Methods” section (Please refer to lines 94-103). 

References

  1. Yan, K.; Wang, X.; Liu, Z.; Bo, Z.; Zhang, C.; Guo, M.; Zhang, X.; Wu, Y. QX-type infectious bronchitis virus infection in roosters can seriously injure the reproductive system and cause sex hormone secretion disorder. Virulence 2023, 14, 2185380, doi:10.1080/21505594.2023.2185380.
  2. Zhang, X.; Yan, K.; Zhang, C.; Guo, M.; Chen, S.; Liao, K.; Bo, Z.; Cao, Y.; Wu, Y. Pathogenicity comparison between QX-type and Mass-type infectious bronchitis virus to different segments of the oviducts in laying phase. Virol J 2022, 19, 62, doi:10.1186/s12985-022-01788-0.

Comment #3: In addition, the use of attenuated vaccines causes other problems in the field - the necessity to differentiate during diagnostics whether we may be dealing with a field or a vaccine strain. Can any changes detected by the authors of this manuscript be used to distinguish vaccine from field strains?

Response: Thank you for your valuable suggestion. We acknowledge the significance of this study. In line with your recommendation, we will conduct a comparative analysis between the attenuated vaccine strain QXL87 and all GI-19 strains, focusing on the mutation sites identified in our research. Our objectives include identifying molecular markers and developing a detection method to differentiate between field strains and vaccine strains. We appreciate your suggestions and anticipate that this endeavor will yield substantial contributions to the field.

Comment #4: I don't know of an IBV strain that affect the nervous system as stated in line 43 - this is probably a mistake.

Response: Thank you. Upon reviewing the literature, it was observed that avian coronaviruses may induce neurological manifestations like encephalitis in chickens. The Furin-S2′ site has been pinpointed as a key element in central nervous system impairment [3,4]. Regrettably, we overlooked including the pertinent literature citation, which has now been rectified in the "References" section (Please refer to lines 422-425).

References

  1. 3. Cheng, J.; Zhao, Y.; Hu, Y.; Zhao, J.; Xue, J.; Zhang, G. The furin-S2' site in avian coronavirus plays a key role in central nervous system damage progression. J Virol 2021, 95, doi:10.1128/jvi.02447-20.
  2. 4. Cheng, J.; Zhao, Y.; Xu, G.; Zhang, K.; Jia, W.; Sun, Y.; Zhao, J.; Xue, J.; Hu, Y.; Zhang, G. The S2 Subunit of QX-type Infectious Bronchitis Coronavirus Spike Protein Is an Essential Determinant of Neurotropism. Viruses 2019, 11, doi:10.3390/v11100972.

Comment #5: I also found inaccuracies in Fig. 1 c - is the scale in this diagram correct?

Response: Thank you for your comment. We regret any lack of clarity in our results explanation. In Figure 1C, the bar graph displays the number of mutations (blue representing synonymous mutations and red representing non-synonymous mutations), while the line graph depicts the gene mutation rate (blue for synonymous mutation rate and red for non-synonymous mutation rate). For example, a non-synonymous mutation is observed in the 3a protein, which spans a gene length of 174bp, resulting in a non-synonymous mutation rate of 0.0057. This description has been added to the legend of Figure 1C, please refer to lines 188-191.
